# Numerical Investigation on the Dynamic Flow Pattern in a New Wastewater Treatment System

**Lubo Tang** [1,2], **Shaohe Zhang** [1,2], **Meng Li** [3], **Xinxin Zhang** [1,2,*] , **Zhanghui Wu** [1,2] **and Like Ma** [1,2]

1 Key Laboratory of Metallogenic Prediction of Nonferrous Metals and Geological Environment Monitoring, Central South University, Ministry of Education, Changsha 410083, China; tanglubo@csu.edu.cn (L.T.); zhangshaohe@163.com (S.Z.); wuzhanghui@csu.edu.cn (Z.W.); marico@csu.edu.cn (L.M.)

2 School of Geosciences and Info-Physics, Central South University, Changsha 410083, China

3 Center for Hydrogeology and Environmental Geology Survey, China Geological Survey, Baoding 071051, China; limeng_sh@mail.cgs.gov.cn

* Correspondence: zhangxx@csu.edu.cn; Tel.: +86-0731-88877811; Fax: +86-0731-88836815

**Abstract:** Currently, industries seek to optimize the development of technology from energy-saving, economic, and environmental perspectives. Dissolved air flotation (DAF) is one of the most effective wastewater treatment systems. However, it requires considerable energy and causes significant operating costs. A recently emerged application of using fluidic oscillators (FOs) to generate microbubbles has attracted extensive attention, as it consumes much less energy and has proven to be a more energy-efficient technique. In this article, a microbubble generator based on FOs is introduced into the flotation tank, and an energy-saving water treatment system, namely fluidic air flotation (FAF), is presented. Using the computational fluid dynamics (CFD) method, the flow pattern of the FAF is investigated. It is observed that FAF generates a dynamic flow pattern, which is beneficial for bubble removal. At the upper part of the separation zone, the flow pattern exhibits a wavy shape. The flow pattern at the lower part switches between clockwise and counterclockwise. The air distribution of the separation zone is also studied. It is found that the height of the "white water" zone almost linearly decreases with the increase in bubble diameter and diffuser size. FAF consumes almost no energy and occupies a small area, and it is expected to provide a promising solution to develop a new generation of the wastewater treatment system.

**Keywords:** computational fluid dynamics; Euler–Euler; fluidic oscillators; microbubble; water treatment

## 1. Introduction

The dissolved air flotation (DAF) system has been extensively used in the wastewater treatment field [1–3]. As shown in Figure 1a, a typical DAF tank consists of two zones separated by a baffle: the contact zone and the separation zone. During the purification processes, air-saturated water under high pressure is injected into the tank by needle valves, and microscopical air bubbles with a diameter of 10–100 μm are formed due to the sudden pressure drop [4]. Bubbles make contact with suspended solids, flocs, or oil droplets [5]. Agglomerates are then formed, which afterward are carried to the separation zone and rise to the surface, where they are removed by scrapers or by overflow. Meanwhile, the purified water is withdrawn from the bottom of this zone through the outlet. The DAF system has received extensive attention in recent decades, and much research has been devoted to developing the theory of flotation and investigating its mechanisms. In a DAF tank, the flow through the separation zone was originally considered plug flows in the vertical direction, and the originally designed surface loading rates of DAF tanks were 6–12 m/h before the 1990s [6]. However, some pilot data were summarized, showing that DAF tanks can be operated at 20–40 m/h, which is much higher than predicted [7,8]. This may be explained by the stratified flow pattern presented by Lundh et al. [9]. The stratified flow was characterized as two different flow patterns (shown in Figure 1b). At

the upper part of the separation zone, a horizontal flow in the direction of the far-end wall was followed by a return horizontal flow immediately below. The lower part with fewer air bubbles demonstrated a plug-like flow. This depicted flow pattern effectively tripled the clarification separation area, which reduced the equivalent clarification hydraulic loading. As a result, the high hydraulic loadings of 20 to 40 m/h can be achieved. In recent years, the computational fluid dynamics (CFD) method has been developed to predict complex multi-phase flow behavior [1]. With such a method, it is possible to obtain further understandings of flow behavior in the DAF process, and great agreement with experiments was achieved. Using a two-dimensional CFD model, Lakghomi et al. [10] concluded that the stratified flow pattern was beneficial for the removal process by increasing the residence time and bubble–bubble contact. Using CFD and population balance equation (PBE), the effect of breakage and coalescence of bubbles on air distribution was studied [11]. It was presented that bubble breakage had a weak influence on the air concentration values. Conversely, the coalescence model had a significant impact on air distribution, but unexpected behavior was observed; namely, the microbubbles did not distribute throughout the height of the separation zone. Behina and Bahramib [4] presented mathematical expressions to model the hydrodynamic characteristics of a DAF tank. Additionally, experimental investigations and CFD simulations were also conducted for comparison, and great agreements were achieved. Although the DAF was a well-established process in the wastewater treatment field, the literature also indicated several obstacles to the scaling up of the DAF system. First, the energy requirement was a significant operating cost because the formation of bubbles depended on the pressurized water [12]. Second, the weather condition affected the efficiency of wastewater treatment; the snow and rain can freeze agglomerates and lead to the settlement phenomenon. Additionally, it was presented that a dead area was observed under the sloping baffle, which had a negative effect on water purification [13,14].

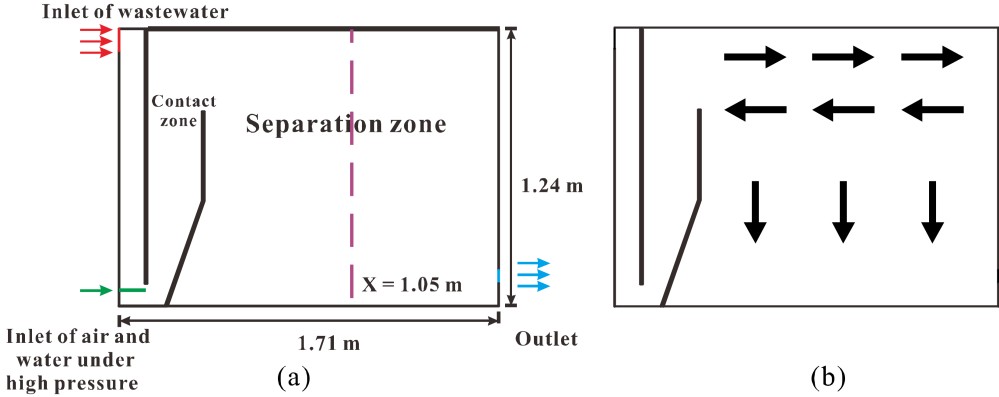

**Figure 1.** (**a**) Geometry of the FAF and DAF tank, (**b**) velocity profile of separation zone in DAF tank.

Another common approach to generate bubbles is to inject air into a diffuser with very small orifices [12]. Unfortunately, the size of bubbles generated this way is around 1 mm, which is much larger than that of DAF [15]. This is because bubbles will be separated from apertures in a diffuser when they are 10 times larger than the diameter of the aperture. Moreover, the polydispersity of bubble sizes and irregularity of the spacing between bubbles also leads to quick coalescence of bubbles. Additionally, the largest bubble formed at apertures provides the path of the least resistance, which is conducive to form larger bubbles. Hence, this method has been only used to break down the major pollutants in wastewater and separate minerals from the host rock in the mining industry [16]. In the last decade, an improved microbubble generator based on the oscillating jet was reported [17]. This device consumes much less energy, occupies a small area, and it is convenient to carry and simple to operate [18]. Tesař [19] analyzed the mechanism of this microbubble generator. They demonstrated that the periodic oscillation of fluidic oscillators (FOs) can prevent the undesirable growth of microbubbles caused by repeated mutual conjunctions

when they are near the exits of the microporous diffuser. As a result, smaller microbubbles were generated, and their size was close to DAF. Hanotu et al. [20] used microbubbles produced by FOs to treat oil-contaminated water. They found that the maximum oil removal efficiency was up to 91%. Based on FOs, Zimmerman et al. [21] designed a bespoke experimental rig that adopted the flexible membrane diffuser network and achieved conventional aeration for a tank with a 30 m$^3$/h throughput. They also illustrated that this kind of microbubble generator is a promising component of wastewater treatment.

In the present study, a new wastewater treatment system composed of an oscillating air jet and DAF tank is presented. It is called fluidic air flotation (FAF) in this paper. In previous research, the airflow rate at the inlet was constant, but the flow rate of FAF oscillated. Its influence on the evolution of flow field and air distribution remains to be revealed. In this paper, CFD transient analysis was used to reveal the complex dynamic fluid flow pattern of FAF. The influence of bubble size and the diameter of the diffuser on the height of the "white water zone" was investigated. The results of this study are anticipated to enrich our understanding of the switching mechanism of FAF and offer references for effectively designing water treatment systems.

## 2. Description of FAF

The bubbles injected into the FAF are generated by the combination of an FO and a diffuser. FOs are capable of creating self-induced periodic oscillating jets without requiring any movable or deformable parts [22,23].

Regarding its application in microbubble generation, the schematic of this device is shown in Figure 2a. The microbubble generator mainly consists of two parts, namely an FO and a diffuser. According to the Coanda effect, the steady airflow supplied from the gas inlet will randomly enter one output channel [24]. If the airflow attaches to the left side and enters the output channel Y1, pressure at the control terminal X1 decreases and draws air through the feedback loop from the control terminal X2. This flow can gain sufficient momentum, which is sufficient to switch the jet to the opposite output channel Y2. By this point, half a cycle has been completed. The switching process is then continued in the same manner, and the self-induced switching process is created. This process produces the oscillating airflow, which will break off from the forming bubble while it is still a hemispherical cap and result in smaller bubbles [17]. Tesař et al. [25] showed that the switching frequency can be adjusted by the flow rate and the length and diameter of the feedback loop. According to the results, when the flow rate was 80 L/min, and the length and diameter of the feedback loop were 4.2 m and 10 mm, respectively, the switching frequency was around 40 Hz, which was conducive to the generation of bubbles [19]. Using the CFD method, the switching process in FO was simulated. As shown in Figure 1b, the gas velocity at one outlet was monitored. The velocity periodically changed, and it was used as the original velocity of the gas inlet of FAF.

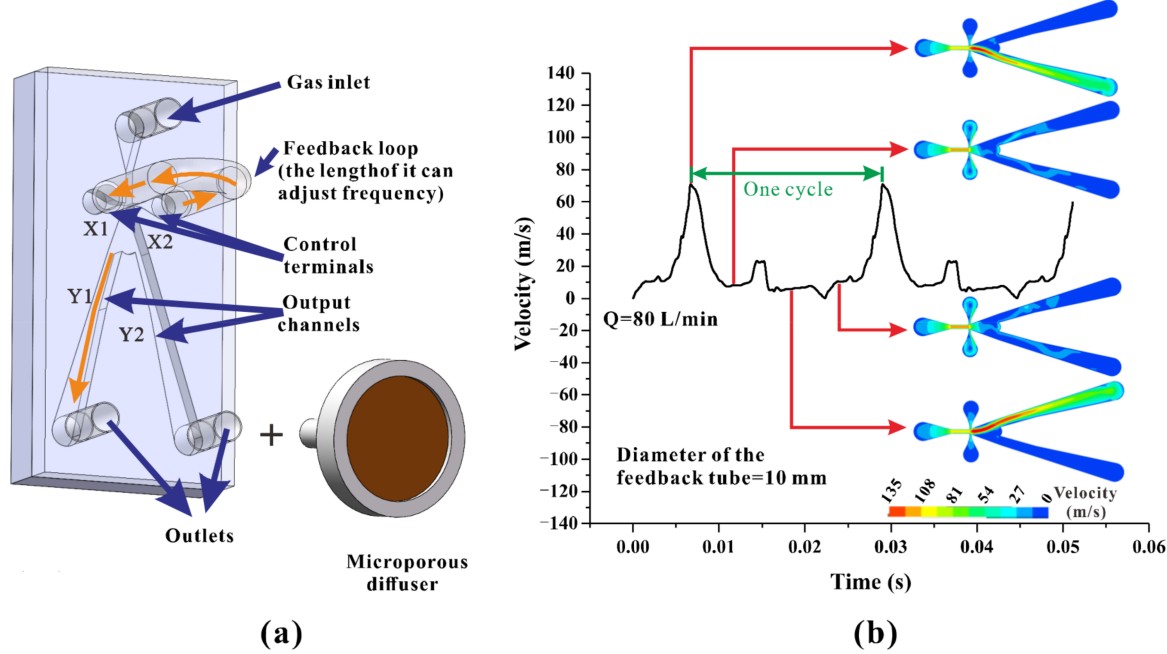

**Figure 2.** (**a**) Schematic of the microbubble generating device composed of fluidic oscillator and microporous diffuser, (**b**) simulated velocity at one outlet of a fluidic oscillator and the corresponding air switching process.

## 3. Numerical Simulations

### 3.1. Computational Domains and Boundary Conditions

In this work, the flotation tank used by Bondelind et al. [26] and Chen et al. [27] was adopted. This tank is 1.71 m long and 1.24 m in height. Bondelind et al. [26] investigated the effects of 2D and 3D models on the flow pattern of DAF, and it was demonstrated that the flow pattern of the separation zone could be successfully reproduced by the 2D model. Additionally, the exclusion of the third dimension significantly decreased the time cost. Because the focus of the present work is to study the flow field and air concentration in the separation zone, the 2D model was a somewhat pragmatic method and was adopted in this work.

The wastewater inlet and the oscillating air inlet were set as the velocity inlet, and the outlet was set as the pressure outlet. The velocity of wastewater was 0.0397 m/s, and that of air was the simulated velocity of the outlet of the fluidic oscillator. Using CFD methods, Rodrigues and Béttega [28] demonstrated that the degassing boundary condition was more suitable than the non-slip wall condition for the water surface. Hence, the degassing boundary condition was used in this study. The main characteristics of the simulations are shown in Table 1.

### 3.2. Time Step and Computational Time

The switching process of airflow in the fluidic oscillator was nearly 0.025 s (shown in Figure 2b), and the time step of 0.002 s was adopted. With this value, around 12 positions of the switching process could be captured, which was thought to be adequate for this work. The computational time of 200 s was adopted as the flow field tended to be steady.

**Table 1.** Numeric parameters and boundary conditions.

| Information | Adopted Condition |
| --- | --- |
| Multiphase model | Euler–Euler |
| Turbulence model | Realizable κ-ε |
| Gravity | 9.81 m/s$^2$ |
| Discretization scheme for the momentum equation | 2nd Order Upwind |
| Discretization scheme for the volume fraction equation | 1st Order Upwind |
| Discretization scheme for the turbulent kinetic energy equation | 2nd Order Upwind |
| Discretization scheme for the turbulence dissipation rate equation | 2nd Order Upwind |
| Average time-step | 0.002 s |
| Total simulated flow time | 200 s |
| Wastewater inlet | Velocity inlet |
| Oscillating air inlet | Velocity inlet |
| Outlet | Pressure outlet |
| Walls and baffles | Wall |
| Surface of flotation tank | Degassing |

### 3.3. Grid Convergence Index

The computational grids were built in the Hypermesh 2019 software (Altair. Engineering Inc., Troy, MI, USA), using a structured mesh. For quantifying the uncertainty of the result due to the grid discretization, the grid independence test was performed using the grid convergence index methodology [28]. Three grid refinements with mesh numbers varying from 2000, 5000, and 8000 cells were tested. Wang et al. [16] showed that the general flow field and mean fluid velocity predictions in the flotation process were not strongly influenced by either the grid resolution or discretization scheme. Considering the acceptable computational time and satisfactory model accuracy, medium grids were chosen.

### 3.4. Turbulence Models and Multiphase Flow

Under the oscillating effect of the fluidic oscillator, the turbulence phenomena in an FAF tank are universal. Additionally, various vortices that exist in the separation zone are closely connected with the working mechanism of FAF. Hence, it is important to choose a proper turbulence model for the internal flow simulation. Lee et al. [1] investigated the influence of turbulence models on the internal flow behavior inside a DAF tank, and it was found that the realizable $k - \varepsilon$ model resulted in greater agreement with the experimental results. This model was also adopted by many researchers [11,26]. Hence, the realizable $k - \varepsilon$ model was used in the following simulations. This model is expressed as

$$\frac{\partial}{\partial t}(\rho k) + \frac{\partial}{\partial x_j}(\rho k u_j) = \frac{\partial}{\partial x_j}\left[\left(\mu + \frac{\mu_t}{\sigma_k}\right)\frac{\partial k}{\partial x_j}\right] + \mu_t S^2 - \rho \varepsilon \tag{1}$$

$$\frac{\partial}{\partial t}(\rho \varepsilon) + \frac{\partial}{\partial x_j}(\rho \varepsilon u_j) = \frac{\partial}{\partial x_j}\left[\left(\mu + \frac{\mu_t}{\sigma_\varepsilon}\right)\frac{\partial \varepsilon}{\partial x_j}\right] + \rho C_1 S_\varepsilon - \rho C_2 \frac{\varepsilon^2}{k + \sqrt{\varepsilon v}} \tag{2}$$

where $k$ is the turbulence kinetic energy, and $\varepsilon$ is the dissipation rate, while $\partial k$ and $\partial \varepsilon$ are the turbulent Prandtl numbers for $k$ and $\varepsilon$. $C_1 = \max\left[0.43, \frac{\eta}{\eta+5}\right]$, $\eta = S\frac{k}{\varepsilon}$, and $S = \sqrt{2S_{ij}S_{ij}}$, and the model constants $C_2 = 1.9$, $\partial k = 1.0$, $\partial \varepsilon = 1.2$.

The selection of the Euler–Euler model and the Euler–Lagrange multiphase model was also discussed by many researchers, and it was demonstrated that the former is more suitable for the simulation of flow pattern in a DAF tank [27,28]. Within the Euler–Euler framework, the continuity equation for phase $q$ can be defined as

$$\frac{\partial}{\partial t}(a_q p_q) + \nabla \cdot (a_q p_q \vec{V}_q) = \sum_{p=1}^{n}(\dot{m}_{pq} - \dot{m}_{qp}) + S_q \tag{3}$$

where $\alpha_q$ denotes the volume fraction of phase $q$, $\rho_q$ denotes the density of phase $q$, $\vec{V}_q$ is the velocity of phase $q$, $\dot{m}_{pq}$ is the mass transfer from phase $p$ to phase $q$ (kg/s), $\dot{m}_{qp}$ is the mass transfer from phase $q$ to phase $p$ (kg/s), $S_q$ denotes the source term. The momentum balance for phase $q$ yields

$$
\begin{aligned}
&\frac{\partial}{\partial t}(a_q p_q \vec{V}_q) + \nabla \cdot (a_q p_q \vec{V}_q \vec{V}_q) \\
&= -a_q \nabla p + \nabla \cdot \overline{\overline{\tau}}_q + a_q p_q \vec{g} + \sum_{p=1}^{n} (\vec{R}_{pq} + \dot{m}_{pq}\vec{V}_{pq} - \dot{m}_q p \vec{V}_{qp}) \\
&\quad + (\vec{F}_q + \vec{F}_{lift,q} + \vec{F}_{wl,q} + \vec{F}_{vm,q} + \vec{F}_{td,q})
\end{aligned}
\tag{4}
$$

where $g$ is the gravitational acceleration (m/s²), $\vec{V}_{pq}$ is interphase velocity (m/s), $\vec{F}_{lift,q}$ is lift force (N), $\vec{F}_q$ is external body force (N), $\vec{F}_{wl,q}$ is wall lubrication force (N), $\vec{F}_{vm,q}$ is virtual mass force (N), $\vec{F}_{td,q}$ is turbulent dispersion force (N), $\overline{\overline{\tau}}_i$ is the stress–strain tensor of phase $q$.

$$
\overline{\overline{\tau}}_q = \alpha_q \mu_q \left( \nabla \vec{V}_q + \nabla \vec{V}_q^T \right) + \alpha_q \left( \lambda_q - \frac{2}{3}\mu_q \right) \nabla \cdot \vec{V}_q \overline{\overline{I}}
\tag{5}
$$

$$
\vec{R}_{pq} = -\vec{R}_{pq}; \; \vec{R}_{qq} = 0
\tag{6}
$$

$$
\sum_{p=1}^{n} \vec{R}_{pq} = \sum_{p=1}^{n} \vec{K}_{pq} \left( \vec{v}_p - \vec{v}_l \right)
\tag{7}
$$

where $\mu_q$ is the viscosity of phase $q$ (Pa·s), $\lambda_q$ is the bulk viscosity of phase $q$ (Pa·s), $\overline{\overline{I}}$ is the deviatoric stress, $\vec{R}_{pq}$ is phase interaction force (N), $\vec{K}_{pq}$ is interphase momentum exchange coefficient, $\vec{v}_p$ is the second phase velocity (m/s), $\vec{v}_l$ is the primary phase velocity (m/s).

## 4. Results and Discussion

The research was concerned with two characteristics of the air flotation system, namely the flow pattern and the air distribution [1,11]. In this section, we systematically analyze the dynamic behavior of flow patterns and air distribution with various parameters.

### 4.1. Analysis of the Dynamic Flow Pattern

FAF generated a new dynamic flow pattern. At the upper part of the FAF, the flow pattern was similar to the stratified flow of the DAF (shown in Figure 3). It was not a complete horizontal flow but exhibited a wavy shape, and this pattern was relatively stable during the purification process. The reason for this flow pattern may be the oscillation effect caused by the fluidic oscillator. The periodic velocity produced vortices in the horizontal layer, and the wavy flow was created. In the lower part of the separation zone, the flow pattern exhibited dynamic characteristics, which switched between clockwise and counterclockwise periodically (shown in Figures 4a and 6a). Additionally, it was observed that when the velocity of the outlet of the fluidic oscillator reached the maximum velocity, the water in the lower part flowed towards the outlet, as shown in Figure 3b. However, this flow pattern was an instant behavior because the peak velocity existed for a very short time. As a result, this flow pattern caused slight damage to the previous flow pattern, but when the peak velocity disappeared, the previous flow pattern was recovered. This instant behavior caused by the oscillation effect of the fluidic oscillator was of great significance to the switching of flow patterns.

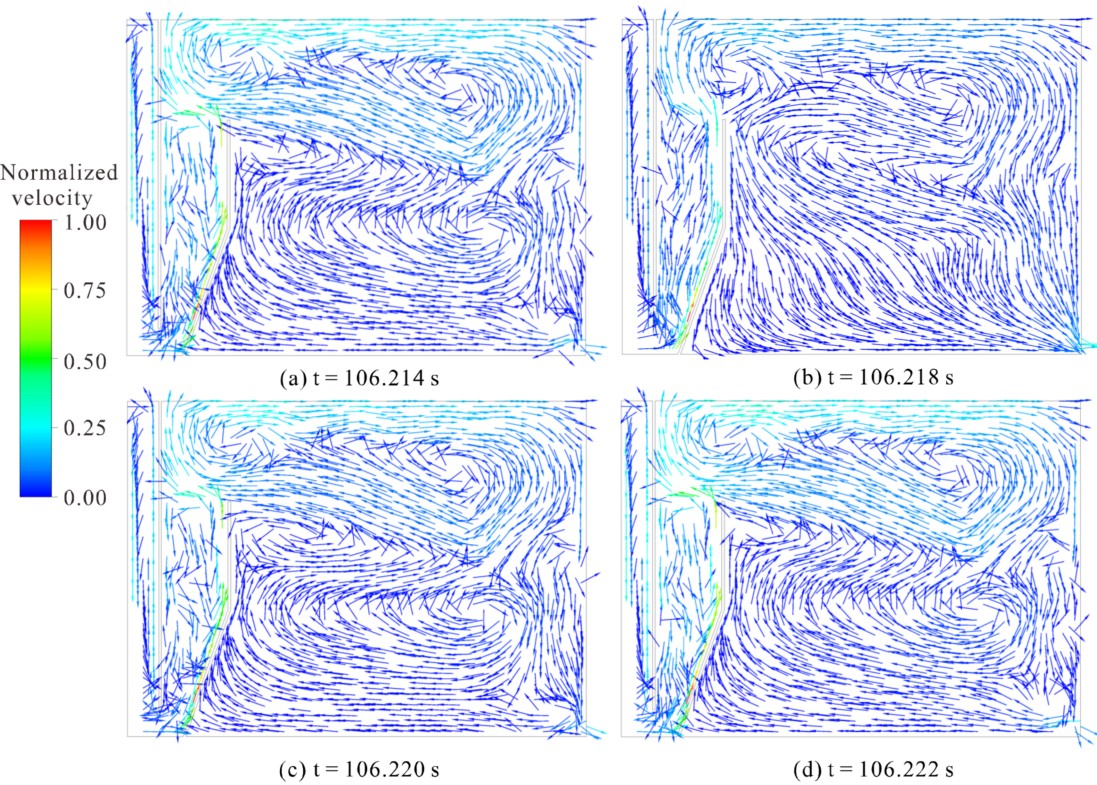

**Figure 3.** Velocity profile before and after the maximum velocity generated by the fluidic oscillator in the FAF tank. (**a**) Velocity profile before the maximum velocity, (**b**) velocity profile at the maximum velocity, and (**c**) and (**d**) velocity profile after the maximum velocity.

Lakghomi et al. [10] showed that the stratified flow of the DAF can help to increase the bubble removal ability by increasing bubble–bubble contact and the residence time of the bubble. This is because larger bubbles, which have larger rise velocities, are formed, and they can be more easily removed. Regarding the flow pattern of FAF, the wavy flow pattern in the upper part may be more effective in increasing bubble–bubble contact and residence time. Additionally, the switching flow pattern in the lower part of the FAF was considered as multiple stratified flows, which was also beneficial in removing the bubble.

*4.2. Switching Mechanism of the Dynamic Flow Pattern*

The dynamic flow pattern in the lower part of the separation zone periodically switched between clockwise and counterclockwise. The switching process from counterclockwise to clockwise is presented in Figure 4. At the beginning of the process, a clockwise vortex was formed on the right side (Figure 3b). This was caused by the larger vertical downward velocity on the right side. As the vortex moved to the left, the vertical downward velocity had little effect on this vortex. It was affected by the horizontal velocity to the left, and the small vortex turned counterclockwise and increased gradually (shown in Figure 4c). In this situation, the contact parts of the two counterclockwise vortices interacted with each other. The flow pattern at the maximum velocity (shown in Figure 3b) also affected the stability of the counterclockwise vortices. As a result, counterclockwise vortices gradually disappeared, as shown in Figure 4d. At this time, the vertical upward velocity on the left was the largest, and a clockwise vortex was formed (shown in Figure 4e). Meanwhile, there was also a small clockwise vortex formed on the right side in the same manner as mentioned above (Figure 4b) because the vertical upward velocity on the left side was greater than the vertical downward velocity on the right side. Finally, the left vortex gradually swallowed the right vortex, and the flow pattern of the lower part switched to the clockwise vortex (shown in Figure 4f). It is worth noting that another switching manner was observed, as shown in Figure 5. If the small clockwise vortex shown in Figure 4b did not move to the

left but moved downward (shown in Figure 5a), the horizontal flow moving to the left was not enough to turn it counterclockwise. Under the influence of the vertical downward flow on the right side, the clockwise vortex became larger, and the counterclockwise vortex gradually disappeared.

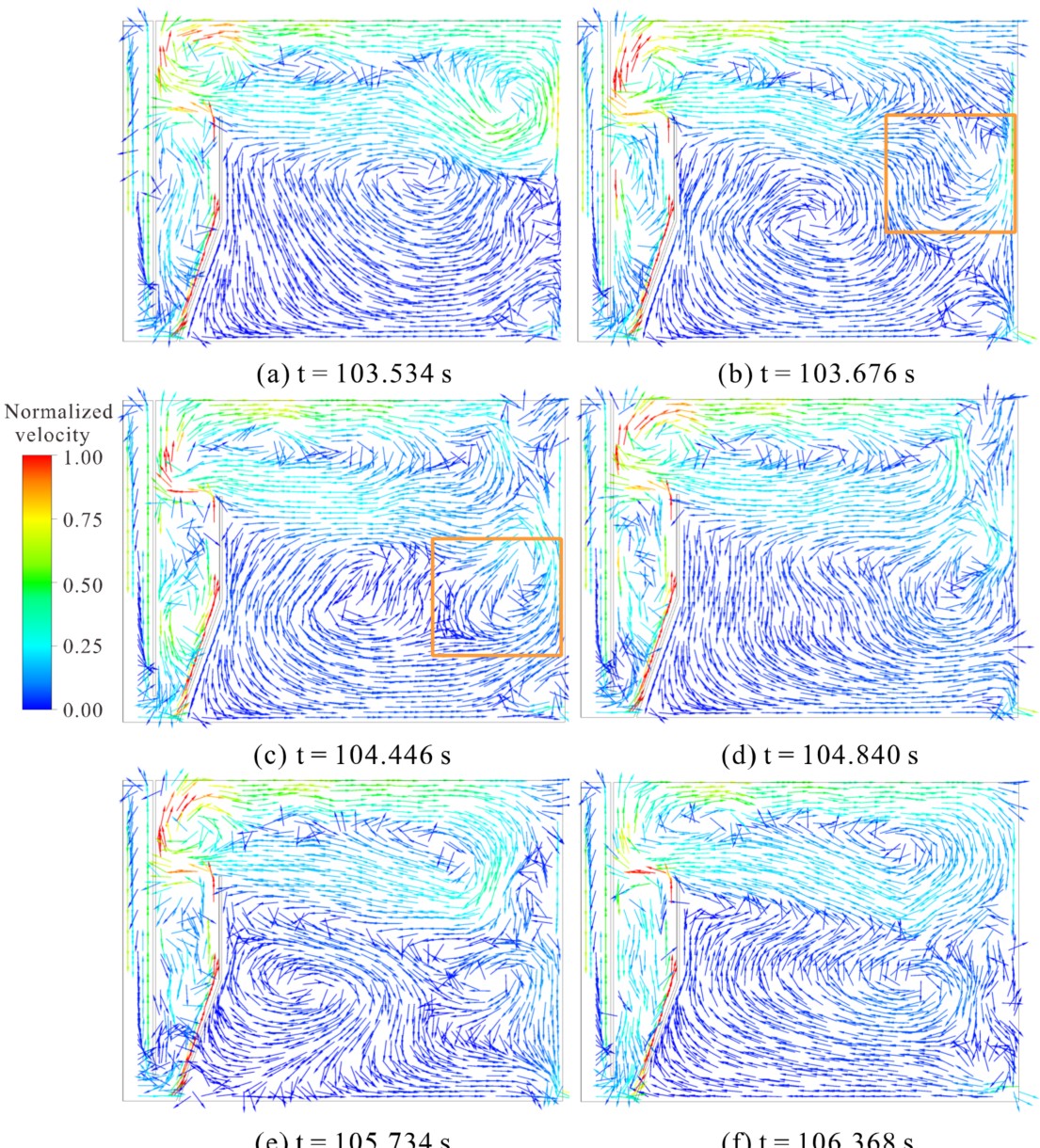

**Figure 4.** Switching process from counterclockwise to clockwise.

The switching process from clockwise to counterclockwise is presented in Figure 6. It was observed that a small counterclockwise vortex was first formed in the top-left direction (shown in Figure 6a). The horizontal flow in the direction to the left and the original clockwise vortex both had positive effects on the generation of this counterclockwise vortex. Hence, the counterclockwise vortex grew easily. Additionally, the periodic flow pattern shown in Figure 3b had a significantly destructive effect on the clockwise vortex. Finally, the counterclockwise vortex replaced the clockwise vortex, and the flow pattern of the lower part switched to counterclockwise again (shown in Figure 6b).

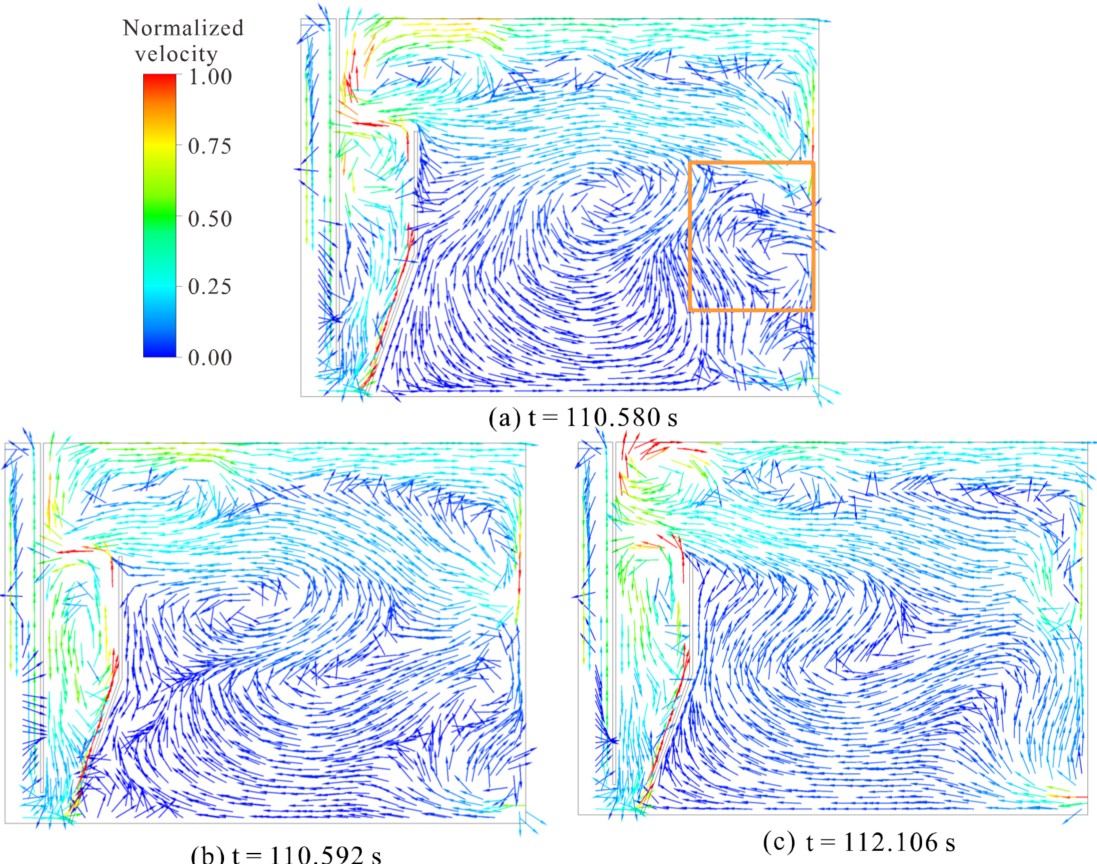

(a) t = 110.580 s

(b) t = 110.592 s

(c) t = 112.106 s

**Figure 5.** Another switching process from counterclockwise to clockwise.

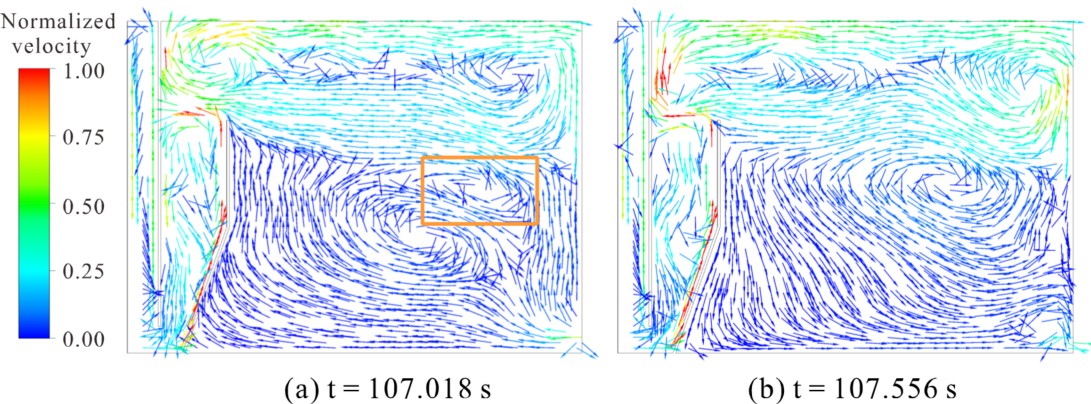

(a) t = 107.018 s

(b) t = 107.556 s

**Figure 6.** Switching process from clockwise to counterclockwise.

In summary, the flow pattern of the FAF was dynamic and complex. The working mechanisms of the FAF were not as simple as they seemed but included the growth and dissipation of the vortex, the coupling between the fluidic oscillator and FAF tank, and the characteristics of multiphase hydrodynamics.

### 4.3. Effect of the Size of Bubble on Air Distribution

A vivid term, "white water zone", has often been used to describe the area where the air concentration is above 1 mL/L [27]. The height of the "white water zone" in the separation zone can intuitively reflect the air distribution. In this section, the height of the "white water zone" in the separation zone under various bubble diameters from 30 to

70 μm was studied. As shown in Figure 7, it was concluded that the height of the "white water zone" decreased almost linearly with the increase in the diameter of the bubble. This is because the rising speed of bubbles in water was proportional to the square of bubble diameter. With the increase in bubble diameter, the rising speed was faster. The air distribution in the upper part of the separation zone also exhibited a wavy shape, and there was a small blank area at the top, both of which were caused by the wavy flow pattern.

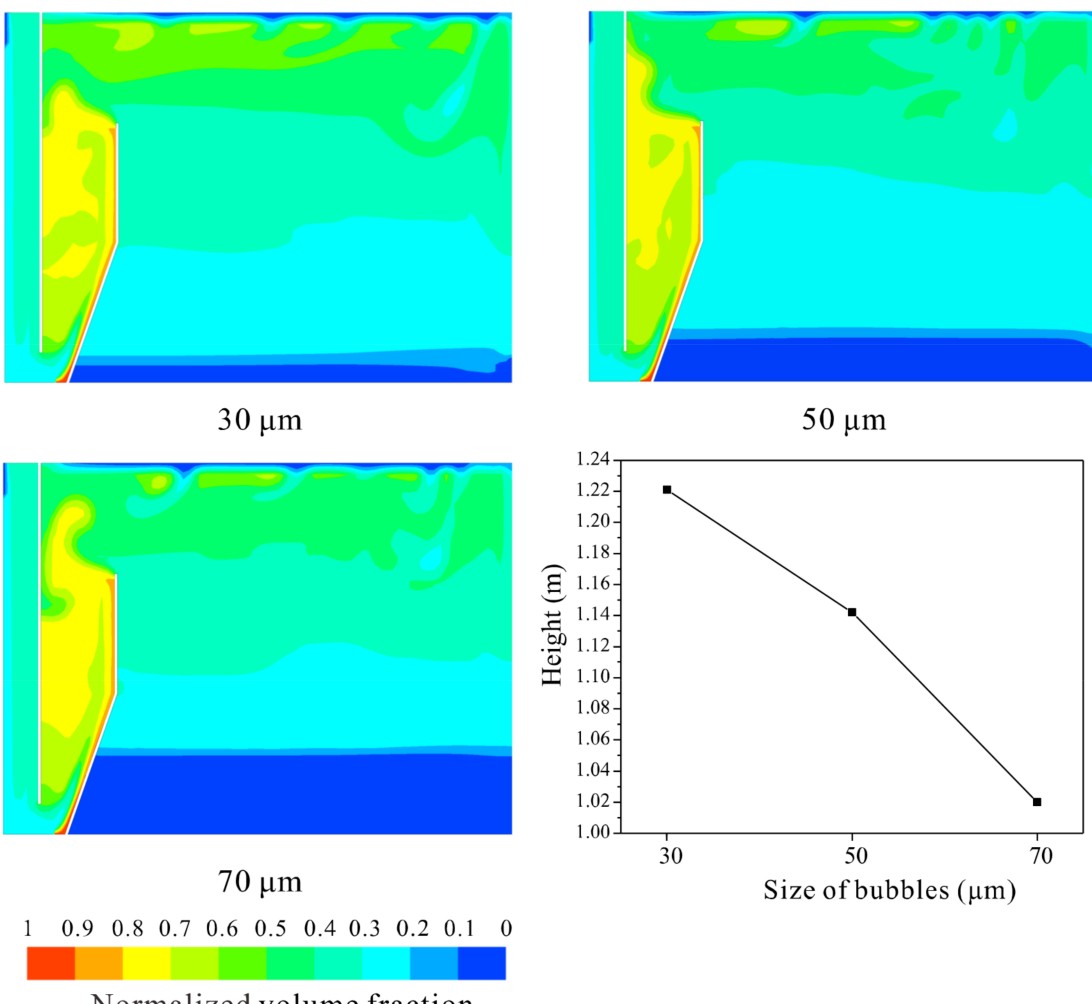

**Figure 7.** Effect of the size of bubble on air distribution.

### 4.4. Effect of the Size of Microporous Diffuser on Air Distribution

The size of the microporous diffuser also affected the air distribution by changing the velocity of the bubbles. As shown in Figure 8, with the increase in the size of the microporous diffuser, the height of the "white water zone" linearly decreased. This can be explained by the fact that under a certain flow rate, with the increase in diffuser size, the velocity of bubbles was lower. Hence, there was not sufficient momentum to force bubbles to move to the bottom of the tank, and more bubbles escaped from the water surface.

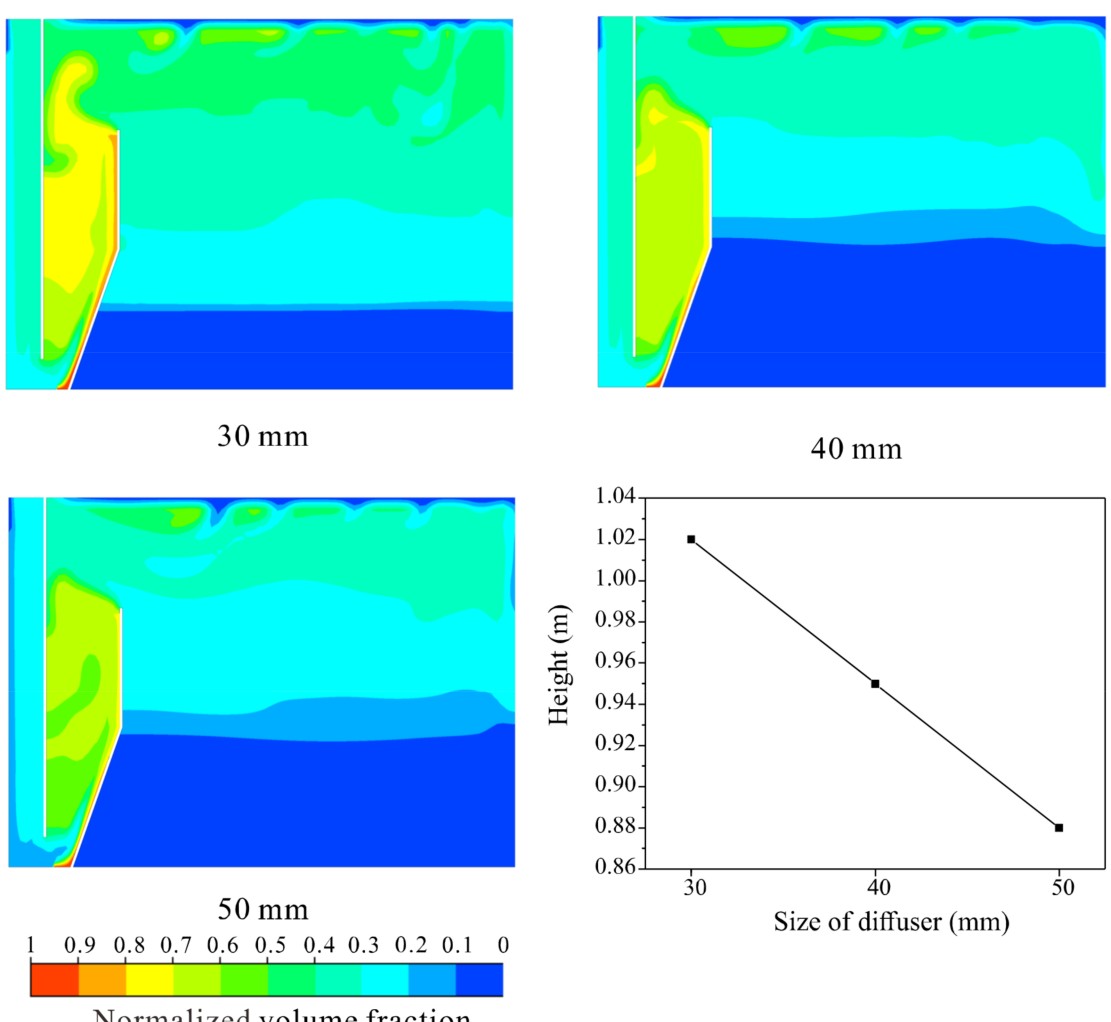

**Figure 8.** Effect of the size of microporous diffuser on air distribution.

## 5. Conclusions

An energy-saving water treatment system, FAF, was presented. Using the CFD method, in this study, we conducted extensive numerical simulations to investigate the dynamic flow pattern and air distribution in FOs. The following findings can be derived from this study.

1. The flow pattern in the separation zone was dynamic. The upper part of the separation zone contained a wavy flow, and the flow pattern at the lower part periodically switched between clockwise and counterclockwise. This dynamic flow pattern can help to improve bubble removal because it leads to the formation of larger bubbles by increasing the residence time and bubble–bubble contact. Additionally, this flow pattern eliminates the dead zone, which also improves the efficiency of wastewater purification.

2. The flow pattern also affected the air distribution, which exhibited a wavy shape in the upper part of the separation zone. The height of the "white water zone" is larger than that of the DAF, which demonstrated that the efficiency of generating bubbles was also improved. It was also found that the height of the "white water zone" almost linearly decreased with the increase in bubble size and microporous diffuser size.

**Author Contributions:** Data curation, L.T. and X.Z.; Formal analysis, L.T. and X.Z.; Investigation, L.T. and X.Z.; Methodology, M.L. and X.Z.; Software, Z.W. and L.M.; Supervision, S.Z. All authors have read and agreed to the published version of the manuscript.

**Funding:** The authors would like to acknowledge the support of the National Natural Science Foundation of China (Grant No. 41872186) and the Natural Science Foundation of Hunan Province (Grant No. 2019JJ50798). The authors also thank the reviewers for their helpful advice.

**Institutional Review Board Statement:** Not applicable.

**Informed Consent Statement:** Not applicable.

**Data Availability Statement:** Not applicable.

**Conflicts of Interest:** The authors declare that they have no known competing financial interests or personal relationships that could have appeared to influence the work reported in this paper.

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
