# Peer review of "Numerical Investigation on the Dynamic Flow Pattern in a New Wastewater Treatment System"

_water, doi:10.3390/w13081101_

Round 1

Reviewer 1 Report

The authors presented the results of numerical investigation on the dynamic flow pattern in a new wastewater treatment system

The paper done by the authors and the outcome is very nice.
The paper is interesting and well written.Overall, the paper need more coherence between results. Some results seems to have no correlation with the others. The analysis of the results should be presented in a more systematic and ranked manner. At times, the results are discussed quite loosely. It needs to be corrected

Other weaknesses to be corrected:
Keywords should be in alphabetical order.
Please improve the manuscript with a English proofreading

I recommend this manuscript for publication in "Water" Journal after completing some minor improvements.

Author Response

Response to Reviewers’ Comments:

Thank you for your letter and the Reviewers’ comments. Based on the comments, I have made an extensive modification to the original manuscript. Here, I attached a revised manuscript with the correction sections marked red. The main corrections in the paper and the response to the comments are as follows.

Reviewer #1: 
This work done by the authors and the outcome is very nice, the paper is interesting and well written.

Point 1: The Reviewer pointed out that the analysis of the results should be presented in a more systematic and ranked manner.
Response 1: The results were revised according to the Reviewer’s comment. 

Point 2: In the second point the Reviewer suggested keywords should be in alphabetical order.
Response 2: The keywords were revised according to the Reviewer’s comment. 

Point 3: In his last comment the Reviewer suggested improving the manuscript with English proofreading. 
Response 3: The manuscript was carefully improved according to the Reviewer’s comment. 

We appreciate for Editors/Reviewers’ warm work earnestly and hope that the correction will meet with approval. Thank you very much for your comments and suggestions.

Best regards,
Xinxin Zhang

Reviewer 2 Report

Dear Authors,

Many thanks for your interesting work. Your paper is really good to show the flow behavior inside such geometry but it is not good from scientific point of view. Please follow the below comments about your research.

  • First of all, you need to be specific and describe all details regarding your research. So, please define all abbreviations prior to their first usage even in abstract like "CFD". So, revise your paper accordingly.
  • I couldnt see the novelty of your work! Would you please describe what is novel and what you are really looking for?
  • CFD is not a Technology it is a computational method and science to help researchers to find better idea regarding their experiments and doing parametric studies and many others...... 
  • Please combine the methodology part and all boundary conditions and time steps in one table! It is more easy to follow! 
  • In figure 2, for the velocity contours you need to provide the dimension and name of it near the colorful legend! 
  • Where is 12 point? Would you please show them! I saw you talked about them but never see that! 
  • Where is your mesh samples! How you did meshing? Number of mesh elements! Structured or unstructured? Mesh types....
  • I couldnt find the validation in your work! Where is your validation for CFD? Where is your validation between CFD and experiments!
  • Where is your mesh sensitivity and mesh convergence index?
  • Please provide a figure to show your mesh details! 
  • Which software did you used? Please provide enough data about that and provide corresponding reference in your reference list! 
  • What is the meaning of dead area? Your CFD results do not show that they are really dead!
  • Please provide more contours Temperature and Pressure! 
  • How you normalized your velocity! What was your reference value! 
  • You need to compare some dimensionless numbers if you are really looking for a good scientific paper! 

My suggestion is resubmit after solving the above-mentioned points. At current form it is not suitable for Water!

Author Response

Response to Reviewers’ Comments:

Thank you for your letter and the Reviewers’ comments. Based on the comments, I have made an extensive modification to the original manuscript.  The main corrections in the paper and the response to the comments are as follows.

Reviewer #2:

This is an interesting study and it is really good to show the flow behavior inside such geometry.

Point 1: The Reviewer pointed out that all abbreviations should be defined when they are used for the first time;

Response 1: Abbreviations were revised according to the Reviewer’s comment.

Point 2: The Reviewer expressed concerns about the novelty of this paper.

Response 2: We presented a new wastewater treatment system, namely, the fluidic air flotation (FAF), by introducing fluidic oscillators into the dissolved air flotation (DAF) tank. Although DAF is one of the most effective wastewater treatment systems, its energy requirement was a significant operating cost because the formation of bubbles depended on the pressurized water. Additionally, the weather condition and dead area of DAF have a negative influence on water purification efficiency.

  However, FAF presented in this paper consumed almost no energy, occupied the little area, and was convenient to carry and simple to operate.  The dynamic flow pattern observed in FAF reduced the dead area and improved the efficiency of purifying wastewater. FAF is expected to provide a promising solution to develop the new generation of the wastewater treatment system.

The characteristics of the air flotation system which researchers concerned were mainly two aspects, namely, the flow pattern and the air distribution. Hence, we studied the flow pattern and the air distribution of FAF.

Point 3: The reviewer pointed out the “CFD technology” should be corrected as “CFD method”

Response 3: It was revised according to the Reviewer’s comment.

Point 4: The reviewer suggests combining the methodology part and all boundary conditions and time steps in one table.

Response 4: This is a very useful comment. Table1 which included the methodology part, boundary conditions, and time steps was given.

Point 5: The reviewer pointed out the dimension and name of the velocity contours in figure 2 should be provided.

Response 5: The dimension and name of the velocity contours in figure 2 were provided.

Point 6: The reviewer expressed concerns about the 12 points.

Response 6: A switching process of airflow in a fluidic oscillator was nearly 0.025 s (shown in Fig. 2b). If the time step was set as 0.002 s, about 12 positions (0.025/0.002) of the flow can be captured in one switching process. And 12 positions are enough to describe the switching process. Hence, 12 positions are used to explain the rationality of time step selection

Point 7: The reviewer expressed concerns about the mesh. For example, mesh types, the software for building mesh, the number of mesh elements, mesh convergence index.

Response 7: As mentioned in this paper, the 2D model was somewhat a pragmatic method and was adopted in this work. This model is very regular and structured mesh is selected. These grids were built in the HyperMesh 2019 software.

The grid convergence index methodology was performed to test grid independence. Wang et al. [1] presented that the general flow field and mean fluid velocity predictions in the flotation process were not strongly influenced by either the grid resolution or discretization scheme. Considering the acceptable computational time and satisfactory model accuracy, medium grids were chosen from three mesh densities with mesh numbers varying from 2,000, 5,000, and 8,000 cells.

Point 8: The reviewer expressed concerns about the validation for CFD.

Response 8: CFD method is quite mature to predict complex multiphase flow behavior. The simulation methodology adopted in this article has been extensively verified by lots scholars in previous relevant studies. It’s feasible to use this method to achieve convincible accuracy results.

Point 9: The reviewer expressed concerns about the dead zone.

Response 9: Dead zone was an area of DAF, which was observed under the sloping baffle. Fluid in this area hardly flows, which reduces the contact between bubbles and pollutants, and has a negative effect on water purification. Dead zone does not exist in fluidic air flotation (FAF), which improves the efficiency of purifying wastewater.

Point 10: The reviewer presented that more temperature and pressure contours should be provided.

Response 10: The characteristics of the air flotation system which researchers concerned was mainly two aspects, namely, the flow pattern and the air distribution. Additionally, the temperature and pressure in FAF change little.

Point 11: The reviewer expressed concerns about the normalized velocity.

Response 11: The normalized velocity was: x' = (x - xmin) / (xmax - xmin). It’s a generally used method for data processing.

Point 12: The reviewer suggested to compare some dimensionless numbers.

Response 12: The characteristics of the air flotation system which researchers concerned was mainly two aspects, namely, the flow pattern and the air distribution. In this paper, we focus on the dynamic flow pattern and the air distribution.

We appreciate for Editors/Reviewers’ warm work earnestly and hope that the correction will meet with approval. Thank you very much for your comments and suggestions.

Best regards,

Xinxin Zhang

  1. Wang, G.; Ge, L.; Mitra, S.; Evans, G.M.; Joshi, J.B.; Chen, S. A review of CFD modelling studies on the flotation process. Miner. Eng. 2018, 127,

Round 2

Reviewer 2 Report

Dear Authors,

Many thanks for your work. The revised version have been improved a lot and it is very close to a scientific paper.